# Effect of Modulation Periods on the Mechanical and Tribological Performance of MoS$_2$–Ti$_L$/MoS$_2$–Ti$_H$ Multilayer Coatings

Ping Zhang [1,2,3], Puyou Ying [1,2,3,*], Changhong Lin [1,2,3], Tao Yang [1,2,3], Jianbo Wu [1,2,3], Min Huang [1,2,3], Tianle Wang [1,2], Yihang Fang [1,2] and Vladimir Levchenko [1,2,3,*]

1   Zhejiang Provincial Key Laboratory for Cutting Tools, Taizhou University, Taizhou 318000, China; zhangp03@tzc.edu.cn (P.Z.); lin201191@163.com (C.L.); yangtaochd@163.com (T.Y.); wujb@tzc.edu.cn (J.W.); mhuang@tzc.edu.cn (M.H.); wtl0203@tzc.edu.cn (T.W.); fangyihang@tzc.edu.cn (Y.F.)
2   School of Pharmaceutical and Materials Engineering, Taizhou University, Taizhou 318000, China
3   Institute of Advanced Coating Materials, Taizhou University, Taizhou 318000, China
*   Correspondence: ypu@tzc.edu.cn (P.Y.); vladalev@yahoo.com (V.L.)

**Abstract:** MoS$_2$–Ti coating is a widely used solid lubricant owing to its low friction coefficient. The mechanical and tribological performance of the coating can be further improved via introducing a multilayer structure, which is closely related to the modulation period and significantly affects the properties of the coating. Herein, the effect of two different modulation periods on the mechanical and tribological performance of the MoS$_2$–Ti$_L$/MoS$_2$–Ti$_H$ multilayer coatings (where L and H represent low and high-powered sputtering of the titanium target) was studied. The performance of the coatings was found to depend on modulation periods of single layer thickness and thickness ratio, respectively. When the thickness ratio of MoS$_2$–Ti$_L$ layer to MoS$_2$–Ti$_H$ layer was fixed with different number of layers, the adverse effects of the interface outweighed the beneficial effect; thus, the mechanical and tribological performance of the multilayer coatings were improved with an increase in the single layer thickness. When the effect of the multilayer interfaces on the studied coatings was similar with the same number of layers, the MoS$_2$–Ti$_H$ layer had more impact on the hardness of the MoS$_2$–Ti$_L$/MoS$_2$–Ti$_H$ multilayer coatings, whereas the MoS$_2$–Ti$_L$ layer substantially affected the adhesion properties, friction behavior and wear resistance. This study can provide a way to regulate coatings with different performance requirements via building different multilayer microstructures.

**Keywords:** modulation periods; MoS$_2$–Ti$_L$/MoS$_2$–Ti$_H$ multilayer coating; single layer thickness; thickness ratio; mechanical and tribological performance

## 1. Introduction

Under certain operating conditions, for wear reduction purposes, solid lubricants are sometimes more suitable than liquid lubricants [1]. For example, in the space industry, solid lubricants are used instead of liquid lubricants because liquid lubricants are unusable at low temperatures—they become too viscous to provide effective lubrication [2]. Solid lubricants are also used instead of liquid lubricants in dry machining operations as using solid lubricants in such applications is more economical and effective [3].

Among the solid lubricants that are commonly used in the industry, sputtered MoS$_2$ coating exhibits an ultra-low friction coefficient [1]. Unfortunately, pure MoS$_2$ coating exhibits low hardness, poor oxidation, moisture and wear resistance, which is related to its porous microstructure. This attribute limits its application [1,4]. In order to solve the above drawbacks, MoS$_2$-based coatings with incorporated additives have been developed, with the resultant doped MoS$_2$-based coatings exhibiting improved performance compared with their undoped counterparts. The doped additives can be classified into four main categories: metals (Ti [5–21], Au [22], Pb [16,20,21,23], Sn [24], Ta [25], W [26], Zr [27],

Ni [28] and Cu [29]), compounds ($Sb_2O_3$ [28], PbO [30], $WS_2$ [31], NbN [32] and TiN [33]), non-metallic elements (B [34], C [35] and N [36]) and composite additives (Mo–S–Ti–C [37], Mo–S–Ti–Pb [38] and $Ti/TiB_2/MoS_2$ [39]).

Of all the dopants, titanium (Ti) is one of the earliest studied and most widely used elements. The mechanical and tribological performance of $MoS_2$–Ti coatings can be remarkably improved owing to the following reasons. First, adding Ti contributes towards the compactness of the coating, thereby reducing the concentration of defects in the material. This is the main reason why such coatings have better mechanical and tribological performance [5–8,10–16,21,38] compared with their undoped counterparts. In contrast, pure $MoS_2$ coatings exhibit a columnar structure, which is porous and loose. Second, incorporating Ti transforms the structure of the $MoS_2$–Ti coating from crystalline to quasi-amorphous/amorphous, which improves the performance of the coating [5–9,13,38]. Third, Ti dissolution into the $MoS_2$ structure causes lattice distortion, generating the solid solution hardening effect. Again, this improves the hardness of the coatings [6–8]. Fourth, $MoS_2$ coatings containing a Ti dopant exhibit inhibition of (001) basal planes and contribute to the presence of (002) basal planes, which enhance friction properties and oxidation resistance of the coatings [5,13,14,20,39]. Fifth, the Ti incorporated into $MoS_2$ coatings readily reacts with water ($H_2O$) and oxygen ($O_2$) to form a dense and protective film, and then slow the oxidation process of the resultant coating. Consequently, $MoS_2$–Ti coatings are more oxidation-resistant than pure $MoS_2$ [5,8,17,21,28,38,39].

Another effective approach to improving the performance of the coating is to build a multilayer microstructure [10–12,19,37]. The multilayer structure effectively inhibits the growth of a column-like structure and makes the coating more compact as the multilayer structure effectively slows dislocation movement and grain boundary slippage [10–12,19]. In addition, the multilayer interface effectively inhibits the infiltration of oxygen elements into the inner coating, thus the oxidation resistance of the coating can be improved [37]. Furthermore, owing to the crack-hindering effects of the interface in coatings, crack formation, crack deflection and crack propagation can be effectively prevented, which reduces the wear of coatings [19,37].

Our previous studies proved that the mechanical and tribological performance of $MoS_2$–Ti coatings could be effectively enhanced using $MoS_2$–$Ti_L$/$MoS_2$–$Ti_H$ multilayer structures [10–12]. The growth of columnar structures could be prevented by the multilayer structure, thus the $MoS_2$–$Ti_L$/$MoS_2$–$Ti_H$ coating had a compact structure and exhibited excellent mechanical and tribological properties. However, our previous studies focused on the comparison between monolayer coating and multilayer coating [10], or the mechanical and tribological properties of just one multilayer structure under different loads [11,12]. The effects of different multilayer structures (such as different single layer thickness and thickness ratio) on the mechanical and tribological performance of the $MoS_2$–$Ti_L$/$MoS_2$–$Ti_H$ multilayer coatings were not involved in our previous studies. It is necessary to investigate the effect of different multilayer structures on the mechanical and tribological performance of the $MoS_2$–$Ti_L$/$MoS_2$–$Ti_H$ coatings to clarify the corresponding relationship between the multilayer structure and the coating performance. Afterwards, the coating performance can be further improved by constructing appropriate multilayer structures. Therefore, herein, the purpose of this study is to investigate the effect of different modulation periods on the mechanical and tribological performance of the $MoS_2$–$Ti_L$/$MoS_2$–$Ti_H$ multilayer coatings, which can provide a way to regulate coatings with different performance requirements via building different multilayer microstructures.

## 2. Materials and Methods

Two substrates were chosen on which to deposit $MoS_2$–$Ti_L$/$MoS_2$–$Ti_H$ multilayer coatings: a monocrystalline Si wafer and a mirror-finished stainless-steel disk. The Si wafer was provided by Zhejiang Lijing Optoelectronic Technology Co., Ltd. (Quzhou, China). It was chosen as a substrate because Si is more easily cut into samples than stainless steel. Thus, it was easy to prepare samples for structural characterization. Chemical

mechanical polishing was used for the Si wafer. The roughness value for the Si wafer was about 0.005 μm. The mirror-finished stainless-steel disk, which had a high surface finish and good surface roughness, was provided by Jiangshan special steel Co., Ltd. (Taizhou, China). It was chosen as a substrate because it was used for mechanical and tribological performance testing and it was not easily broken in the relevant tests. The polishing procedure for the stainless steel was mechanical polishing, which contained a series of grinding and polishing processes, including rough grinding, fine grinding and ultra-fine grinding. The roughness value for the stainless-steel disk was about 0.01 μm.

In each case, the coatings were deposited via magnetron sputtering (JGP560B1, Sky Technology Development Co., Ltd., Shenyang, China), for which two targets were adopted. One of the two targets was a Ti (99.9%) plate and the other one was a $MoS_2$ (99.9%) plate. The Ti target was connected to direct-current (DC) sputtering power and the $MoS_2$ target was connected to radio frequency (RF) sputtering power. The size of the cylindric targets was φ 50.8 mm × 5 mm, and the distance between the substrate and target was 180 mm. The schematic diagrams of the deposition system are shown in Figure 1. Argon (99.9%) was chosen as the working gas. Before depositing the multilayer coating, an ~100 nm-thick layer of Ti was pre-deposited as an interlayer. The different multilayer structure of $MoS_2$–$Ti_L$/$MoS_2$–$Ti_H$ coatings could be achieved via changing modulation periods such as single layer thickness and thickness ratio, which can be achieved by changing the power of the Ti target and the deposition time of different powers. High-powered sputtering of the Ti target implied that the deposited coating had high Ti content. Similarly, low-powered sputtering of the Ti target implied that the deposited coating had low Ti content. The deposition time under a given power determined the layer thickness. Based on our previous work [11,12], 50 W and 100 W were selected as low- and high-powered sputtering of Ti target, and deposition times of 300 s, 600 s and 900 s were selected to reflect the different single layer thicknesses of the coatings. Therefore, by the combination of different Ti target power and deposition time, different multilayer structures could be obtained.

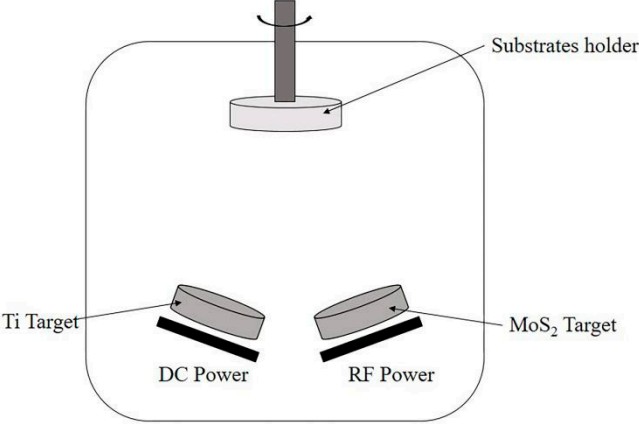

**Figure 1.** The schematic diagrams of the deposition system.

Samples were named in the following way. In the case of sample S300-900, the labelling indicates that the deposition time of the Ti target with different power was alternated between 50 W/300 s and 100 W/900 s while the deposition time of the $MoS_2$ target was maintained at 100 W/1200 s. The total deposition time for each sample was 4 h. Similarly, by adjusting the sputtering time of the Ti target with different power (50 W and 100 W), other samples were obtained. The naming and relevant deposition parameters of all samples are shown in Tables 1 and 2.

**Table 1.** Related parameters for preparing $MoS_2$–$Ti_L$/$MoS_2$–$Ti_H$ coating with different single layer thickness.

| Samples | Deposition Time of $MoS_2$–$Ti_L$ Layer (s) | Deposition Time of $MoS_2$–$Ti_H$ Layer (s) | Total Layers of the Coating | Total Deposition Time (h) |
|---|---|---|---|---|
| S300-300 | 300 | 300 | 24 | 4 |
| S600-600 | 600 | 600 | 12 | 4 |
| S900-900 | 900 | 900 | 8 | 4 |

**Table 2.** Related parameters for preparing $MoS_2$–$Ti_L$/$MoS_2$–$Ti_H$ coating of 12 layers with different thickness ratio of single layer.

| Samples | Deposition Time of $MoS_2$–$Ti_L$ Layer (s) | Deposition Time of $MoS_2$–$Ti_H$ Layer (s) | Total Layers of the Coating | Total Deposition Time (h) |
|---|---|---|---|---|
| S300-900 | 300 | 900 | 12 | 4 |
| S600-600 | 600 | 600 | 12 | 4 |
| S900-300 | 900 | 300 | 12 | 4 |

The deposition rate of 4 h indicated in Tables 1 and 2 do not seem to be very realistic from an industrial production point of view. The magnetron sputtering system in this study is a common balanced magnetron sputtering for scientific research. The size ($\varphi$ 50.8 mm $\times$ 5 mm), number (a total of 3) and power ($\leq$100 W) of target are limited. In order to shorten the deposition time, the methods such as increasing the target size, increasing the target power and co-sputtering with more targets can be adopted. The deposition rate affects structures and properties of the coating, such as adhesion, internal stress, hardness, surface roughness, surface morphology, and so on [40]. Another effective way to shorten the deposition time is to adopt other magnetron sputtering systems such as unbalanced magnetron sputtering. It can overcome some drawbacks of balanced magnetron sputtering, such as low deposition rates and low ionization efficiencies in the plasma [40]. The 4 h deposition time adopted in the present study is because of equipment limitations and to control the thickness and performance of the coatings.

A qualitative study of coating adhesion was conducted by Rockwell-C indentation with a load of 1470 N (150 kgf). The indenter is a diamond indenter, which is a cone with a conical angle of 120°. According to this HRC-DB test [41], the adhesion quality of coatings can be distinguished. Rockwell-C indentation tests were performed three times for each sample.

A Nano test Vantage (Micro Materials, Wrexham, UK) instrument was adopted to test the hardness (H) and elastic (E) moduli of the $MoS_2$–$Ti_L$/$MoS_2$–$Ti_H$ multilayer coatings. The experimental details for the nanoindentation are as follows. It was a dynamic indentation with a Berkovich diamond indenter. Indentations were load controlled at 3 mN maximum load. This maximum load was taken to ensure that the penetration depth of testing coatings was below 120 nm (around 1/7–1/10 of the film thickness), where the contribution of the steel substrate to the results could be ignored. The loading rate was the same as the unloading rate, both of which were 0.1 mN/s. The holding period at peak load was 30 s. The Oliver and Phar theory was used for the calculation of H and E [42]. When calculating the value of *E*, the Poisson's ratios of the indenter and coatings are 0.07 and 0.18, respectively. Three indentations were done for each sample to give an error to each H and E value.

The friction and wear behavior testing was conducted on a reciprocating friction and wear testing machine (UMT-3, Bruker Nano Surfaces Division, Campbell, CA, USA) for 20 min, wherein the applied loads were 5, 10, 15 and 20 N. The experiments were conducted at room temperature under a relative humidity of ~60%. The sliding distance L was 5 mm and the sliding velocity was 20 mm/s. Tests were conducted using a GCr15 steel ball, the

radius, $R_0$, of which was 1.5 mm. Three friction tests of each sample were performed in order to reduce experimental error. Based on the Hertzian contact theory [43], the Hertzian contact of the friction and wear behavior testing is a contact of a ball and a plane, thus the initial Hertzian contact pressure F can be calculated by the equation

$$F^3 = \frac{6}{\pi^3} \times \frac{1}{R_0^2} \times \frac{P}{\left( \frac{1-\mu_1^2}{E_1} + \frac{1-\mu_2^2}{E_2} \right)^2} \tag{1}$$

where $R_0$ is the radius of spherical indenter, $P$ is the applied load and $\mu_1$ and $E_1$ are Poisson's ratio and elastic modulus of the spherical indenter, which are 0.3 and 206 GPa, respectively. $\mu_2$ and $E_2$ are Poisson's ratio and elastic modulus of the test coating, respectively. Based on Equation (1), the initial Hertzian contact pressure, $F$, in the friction and wear behavior testing can be calculated. The initial Hertzian contact pressures under loads of 5, 10, 15 and 20 N were about 1.50, 1.88, 2.16 and 2.37 GPa, respectively. Those values above of initial Hertzian contact pressures indicate high contact pressure, which can represent some real applications such as the mechanical components deposited with solid lubricants (such as $MoS_2$–Ti coatings) have contact and relative motion under high-load conditions [44].

The surface and cross-sectional morphologies of the multilayer coatings were observed via scanning electron microscopy (SEM, S4800, Hitachi, Tokyo, Japan). The coating thickness was the average value of the ten measurements. The wear traces were examined via confocal microscopy (CSM700, Axio, Carl Zeiss, Oberkochen, Germany). The wear rate, W, was calculated by the equation [14]

$$W = V/(F \cdot L) \tag{2}$$

where $V$ represents the volume loss of the coatings, which can be estimated according the cross-sectional profiles of the wear trace computed using Origin software; $F$ represents the applied load and $L$ represents the sliding distance. Three different wear traces were examined for each sample to give an error to the measured wear trace width and the calculated wear rate. The wear resistance of the coatings was estimated from the calculated wear rate.

Analysis of variance (ANOVA) was adopted to statistically recognize the effect of deposition parameters and test loads on the wear trace width and wear rates of the samples. The procedure used for ANOVA technique can refer to Table 4 in reference [45]. ANOVA was carried out for significance level $\alpha = 0.05$, means $1 - \alpha = 0.95$ for a confidence level of 95%. The hypothesis statements are as follows. Null hypothesis ($H_0$): the samples have the same wear trace width and wear rates for various samples and applied loads. Alternative hypothesis ($H_a$): the samples do not have the same wear trace width and wear rates for various samples and applied loads.

## 3. Results

### 3.1. Effect of Single Layer Thickness on the Properties of $MoS_2$–$Ti_L$/$MoS_2$–$Ti_H$ Multilayer Coatings (S300-300, S600-600 and S900-900)

3.1.1. Structure Characterization of $MoS_2$–$Ti_L$/$MoS_2$–$Ti_H$ Multilayer Coatings with Different Single Layer Thickness

Surface and cross-sectional morphologies of the $MoS_2$–$Ti_L$/$MoS_2$–$Ti_H$ multilayer coatings with different single layer thicknesses are presented in Figures 2 and 3. As shown in Figure 2a–c, the surfaces of all the coatings were smooth and dense, with no significant difference in the surface topography between the three images. The cross-sectional morphologies in Figure 3 all showed multilayer structures with alternating layers. The numbers of alternating layers in Figure 3a–c were 24, 12 and 8, respectively. Each alternating layer consisted of a $MoS_2$–$Ti_L$ layer and a $MoS_2$–$Ti_H$ layer, which was not evident in Figure 3. The thicknesses of the S300-300, S600-600 and S900-900 coatings shown in Figure 3 were $0.84 \pm 0.01$, $1.26 \pm 0.01$ and $1.08 \pm 0.01$ μm, respectively. Because of the multilayer structure

of the coatings, the columnar growth of all coatings was inhibited [10–12,19], which is consistent with the results shown in Figure 3.

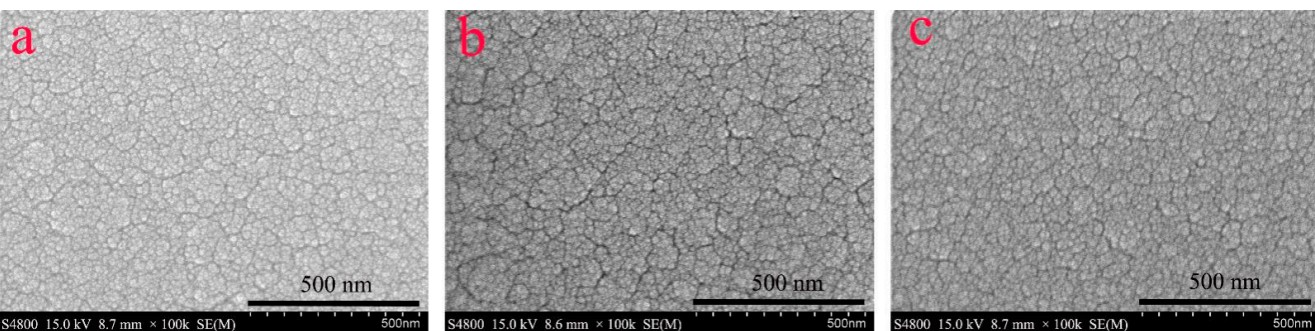

**Figure 2.** SEM surface images of $MoS_2$–$Ti_L$/$MoS_2$–$Ti_H$ coatings: (**a**) sample S300-300; (**b**) sample S600-600; (**c**) sample S900-900.

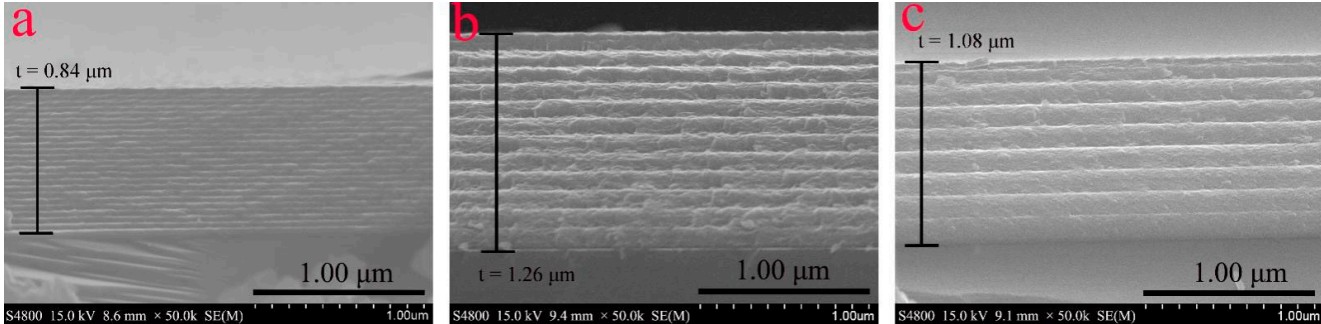

**Figure 3.** Cross-sectional SEM images of $MoS_2$–$Ti_L$/$MoS_2$–$Ti_H$ coatings: (**a**) sample S300-300; (**b**) sample S600-600; (**c**) sample S900-900.

### 3.1.2. Adhesion Properties of $MoS_2$–$Ti_L$/$MoS_2$–$Ti_H$ Multilayer Coatings with Different Single Layer Thickness on the Steel Substrate

Adhesion properties of $MoS_2$–$Ti_L$/$MoS_2$–Ti multilayer coatings with different single layer thicknesses on the steel substrate were evaluated via Rockwell-C indentation. The metallographic images of the indentation craters after the test are shown in Figure 4. It is clearly observable that the S300-300 coating exhibited a large area of coating spallation after indentation. The peeled area was much smaller in the case of the S600-600 coating. It is obvious that the S900-900 coating exhibited excellent adhesion because no spallation occurred after indentation. It can be concluded from Figure 4 that the adhesion properties of the three coatings were found to be in the order of: S900-900 > S600-600 > S300-300, which indicate that the adhesion properties deteriorated with the decrease of single layer thickness.

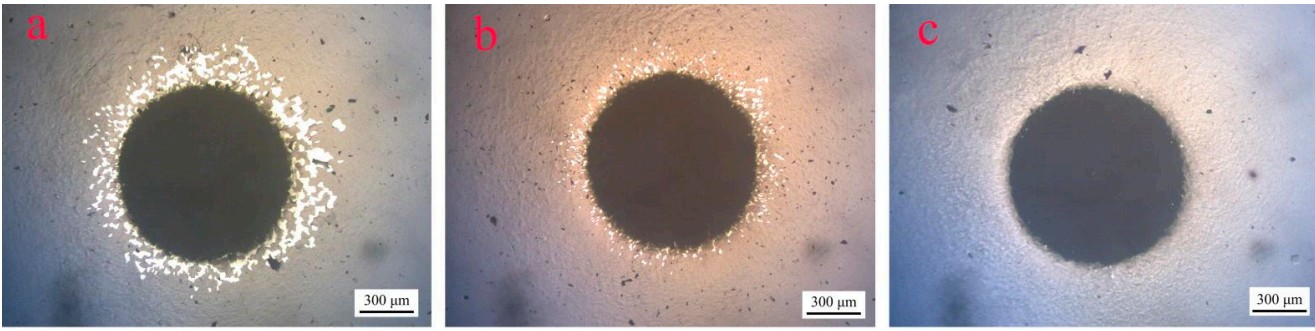

**Figure 4.** Optical micrograph of $MoS_2$–$Ti_L$/$MoS_2$–$Ti_H$ coatings after Rockwell-C indentation: (**a**) sample S300-300; (**b**) sample S600-600; (**c**) sample S900-900.

### 3.1.3. Mechanical Properties of MoS$_2$–Ti$_L$/MoS$_2$–Ti$_H$ Multilayer Coatings with Different Single Layer Thickness

The hardness values and elastic moduli of the multilayer coatings with different single layer thicknesses are listed in Table 3. For the coatings wherein the single layer deposition time increased from 300 to 900 s, the hardness increased from 7.6 to 8.6 GPa. In addition, the elastic moduli showed the same trend as that of the hardness values. The H/E and H$^3$/E$^2$ ratios are presented in Table 3, wherein it can be observed that the H/E ratios of the three samples were close. As single layer deposition time increased, the H$^3$/E$^2$ ratios slightly increased. The results in Table 3 indicate that the mechanical properties deteriorated with the decrease of single layer thickness.

**Table 3.** Hardness values and elastic moduli of MoS$_2$–Ti$_L$/MoS$_2$–Ti$_H$ coatings with different single layer thickness.

| Samples | Total Layers | Hardness (GPa) | Elastic Moduli (GPa) | H/E | H$^3$/E$^2$ |
| --- | --- | --- | --- | --- | --- |
| S300-300 | 24 | 7.6 $\pm$ 0.5 | 131.5 $\pm$ 12.8 | 0.057 $\pm$ 0.002 | 0.025 $\pm$ 0.001 |
| S600-600 | 12 | 8.2 $\pm$ 0.5 | 144.3 $\pm$ 9.3 | 0.057 $\pm$ 0.001 | 0.027 $\pm$ 0.001 |
| S900-900 | 8 | 8.6 $\pm$ 0.4 | 150.1 $\pm$ 6.2 | 0.057 $\pm$ 0.001 | 0.028 $\pm$ 0.002 |

### 3.1.4. Tribological Properties of MoS$_2$–Ti$_L$/MoS$_2$–Ti$_H$ Multilayer Coatings with Different Single Layer Thickness

Figure 5 shows the friction coefficient dynamics of the MoS$_2$–Ti$_L$/MoS$_2$–Ti$_H$ multilayer coatings with different single layer thickness under applied loads of 5–20 N. The friction coefficients of the S600-600 and S900-900 coatings remained stable and low (~0.05–0.1) under all loads, confirming the excellent tribological performance of the S600-600 and S900-900 coatings. However, the tribological performance of the S300-300 coatings deteriorated when the applied load increased from 5 to 20 N. As shown in Figure 5a, the friction coefficient of the S300-300 coating suddenly increased above 0.1 at ~1100 s under the applied load of 5 N. When the applied load increased to 10 N (Figure 5b), the friction coefficient suddenly increased at an earlier time (850 s). Under 15 N and 20 N loads, the friction coefficient was initially low, but suddenly increased at earlier times of 450 and 200 s, respectively.

Figure 6 shows images of the wear trace morphologies of the MoS$_2$–Ti$_L$/MoS$_2$–Ti$_H$ multilayer coatings with different single layer thicknesses under different loads. After 20 min friction test under a load of 5 N, the S300-300 coating still existed on the substrate; however, the surface of the coating started to deform at the wear-trace edge. The deformation of the coating surface at the wear-trace edge was more severe when the applied load increased to ≥10 N, as depicted by the red rectangles in Figure 6d,g,j. Moreover, when tested under large loads, the coating peeled off from the wear-trace area, resulting in rapid lubrication loss of the coating. These observations showed that the wear resistance of the S300-300 coating was poor when the applied load was ≥10N. The surface of the S600-600 coating started to deform at the wear-trace edges under 15 N and 20 N loads, as depicted by the red rectangles in Figure 6h,k. The S900-900 coating was firmly adhered to the substrate, and the surface slightly deformed at the wear-trace edges. The S900-900 coating fully utilized its lubrication under these loads.

The wear trace widths of the multilayer coatings with different single layer thickness (corresponding to those in Figure 6) are listed in Table 4. As shown in Table 4, the wear trace widths of all coatings increased with increasing applied loads from 5 to 20 N. Under each test load, the wear trace width of the S900-900 coating was the smallest, followed by that of the S600-600 coating, with the largest being that of the S300-300 coating. When the applied load was 5 N, there was little difference in the wear trace widths of the S300-300 and S600-600 coatings, which were a little larger than that of the S900-900 coating. This indicated the excellent wear resistance of the three coatings under a small load (5 N). When the applied load increased to ≥10 N, the wear trace width of the S600-600 coating remained

a little larger than that of the S900-900 coating. However, the wear trace width of the S300-300 coating considerably increased, making it much larger than those of the S600-600 and S900-900 coatings. This indicated that the wear resistance of the S300-300 coating was poor, whereas the S900-900 coating maintained its excellent wear resistance under large loads ($\geq$10 N).

Based on the results shown in Figure 6 and Table 4, the S900-900 and S300-300 coatings exhibit the best and worst wear resistances, respectively, which is in accordance with the friction test results shown in Figure 5.

**Table 4.** Wear trace width (μm) of $MoS_2$–$Ti_L$/$MoS_2$–$Ti_H$ coatings with different single layer thicknesses in Figure 6.

| Samples | Loads | | | |
|---|---|---|---|---|
| | 5 N | 10 N | 15 N | 20 N |
| S300-300 | $238 \pm 8$ | $480 \pm 10$ | $834 \pm 14$ | $861 \pm 18$ |
| S600-600 | $214 \pm 9$ | $250 \pm 5$ | $391 \pm 7$ | $443 \pm 9$ |
| S900-900 | $178 \pm 5$ | $234 \pm 3$ | $266 \pm 4$ | $307 \pm 3$ |

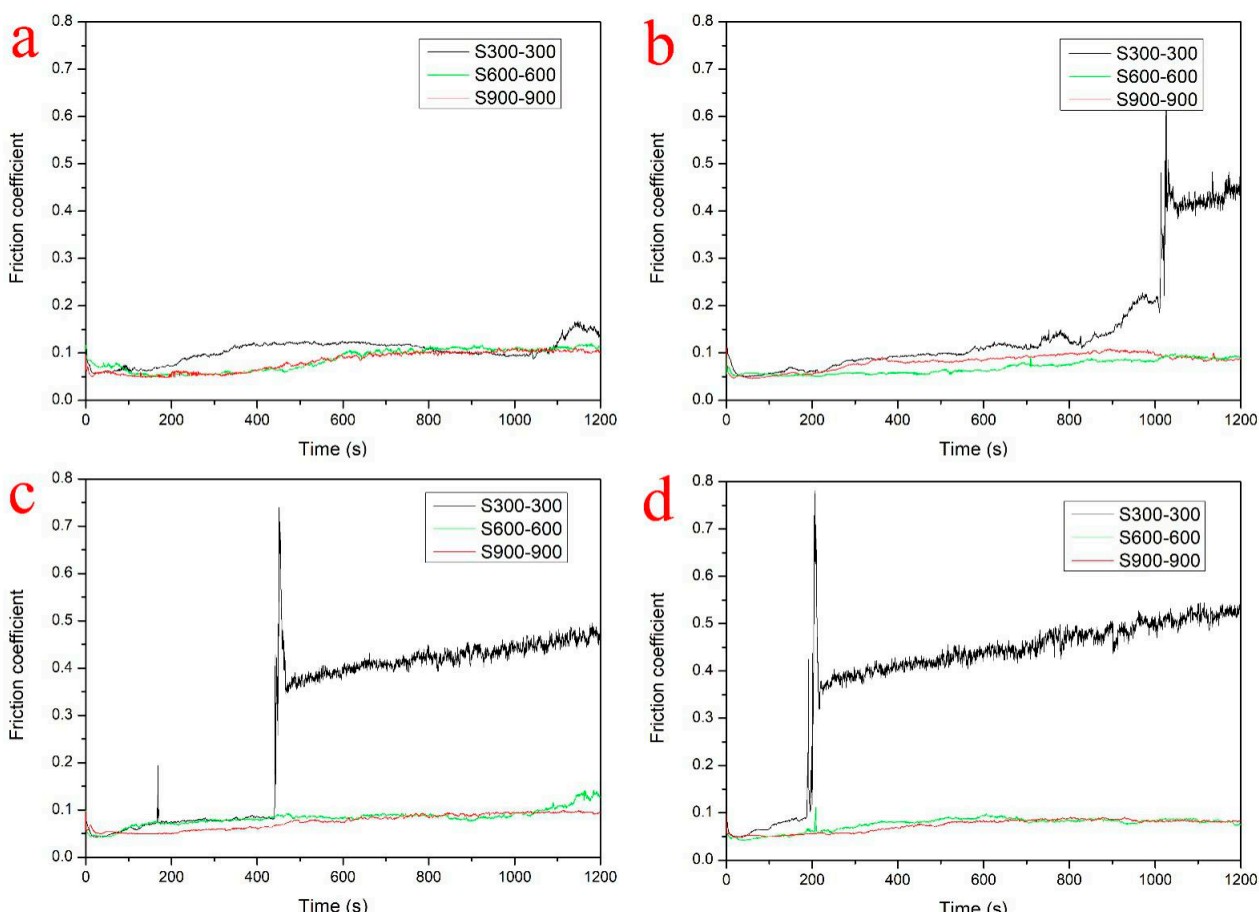

**Figure 5.** Friction coefficient dynamics of the $MoS_2$–$Ti_L$/$MoS_2$–$Ti_H$ coatings with different single layer thickness under different loads: (**a**) 5 N; (**b**) 10 N; (**c**) 15 N; (**d**) 20 N.

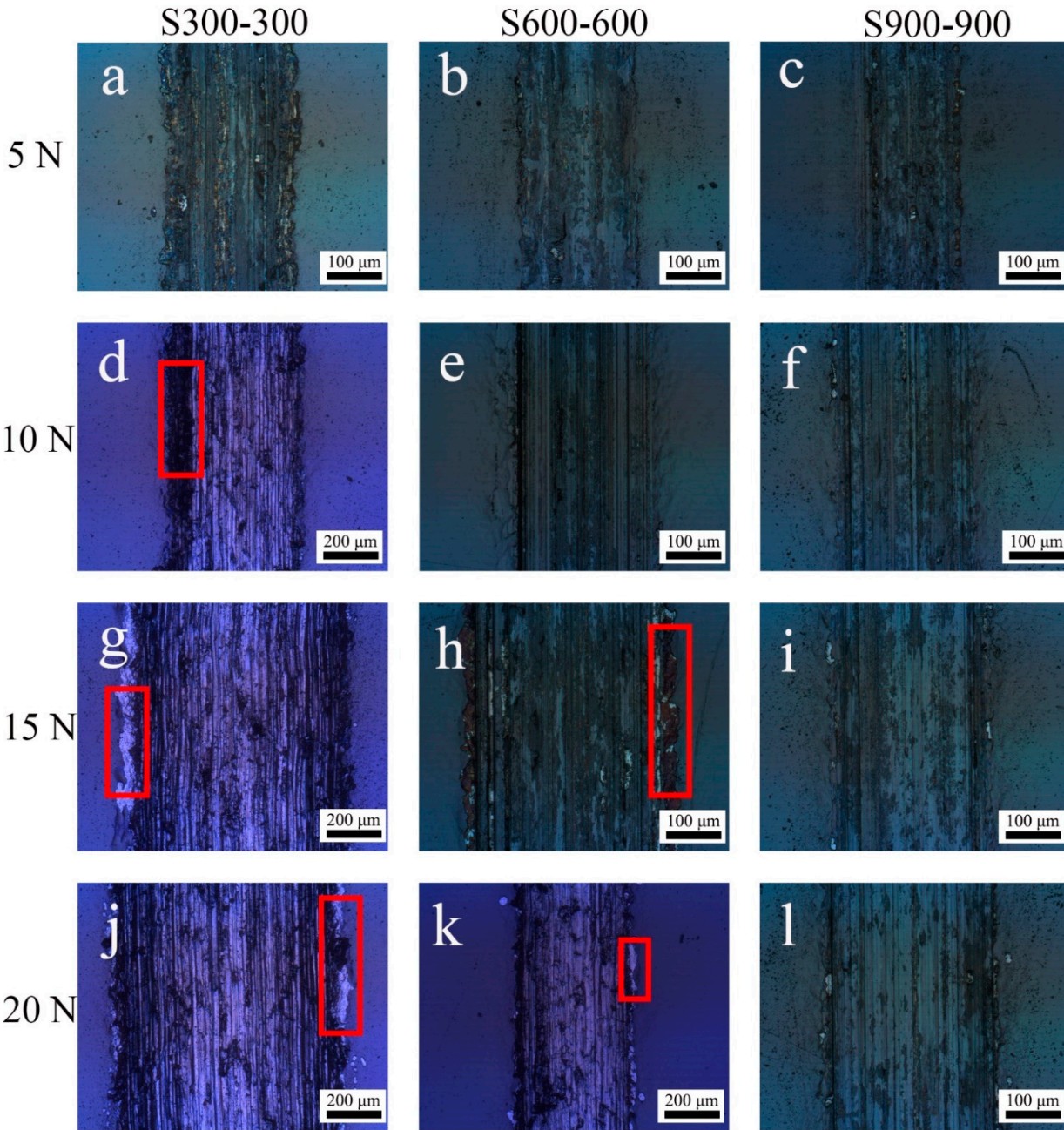

**Figure 6.** Wear trace morphology of the $MoS_2$–$Ti_L$/$MoS_2$–$Ti_H$ coating with different single layer thicknesses under different loads. Sample S300-300 under different loads: (**a**) 5 N, (**d**) 10 N, (**g**) 15 N, (**j**) 20 N; sample S600-600 under different loads: (**b**) 5 N, (**e**) 10 N, (**h**) 15 N, (**k**) 20 N; sample S900-900 under different loads: (**c**) 5 N, (**f**) 10 N, (**i**) 15 N, (**l**) 20 N.

Figure 7 shows the cross-sections of the wear traces of the $MoS_2$–$Ti_L$/$MoS_2$–$Ti_H$ multilayer coatings with different single layer thicknesses under different loads. As shown in Figure 7, the wear trace under 5 N was considerably shallower on all coatings than that under other loads. Under loads of $\geq$10 N, the wear loss of the S600-600 and S900-900 coatings was considerably lower than that of the S300-300 coating. Although the grooves of all coatings deepened as the applied loads increased, the wear traces of the S600-600 and S900-900 coatings were shallow, whereas the wear trace of the S300-300 coating was extremely deep under loads of $\geq$10 N, again confirming that the S300-300 coating exhibited the worst wear resistance.

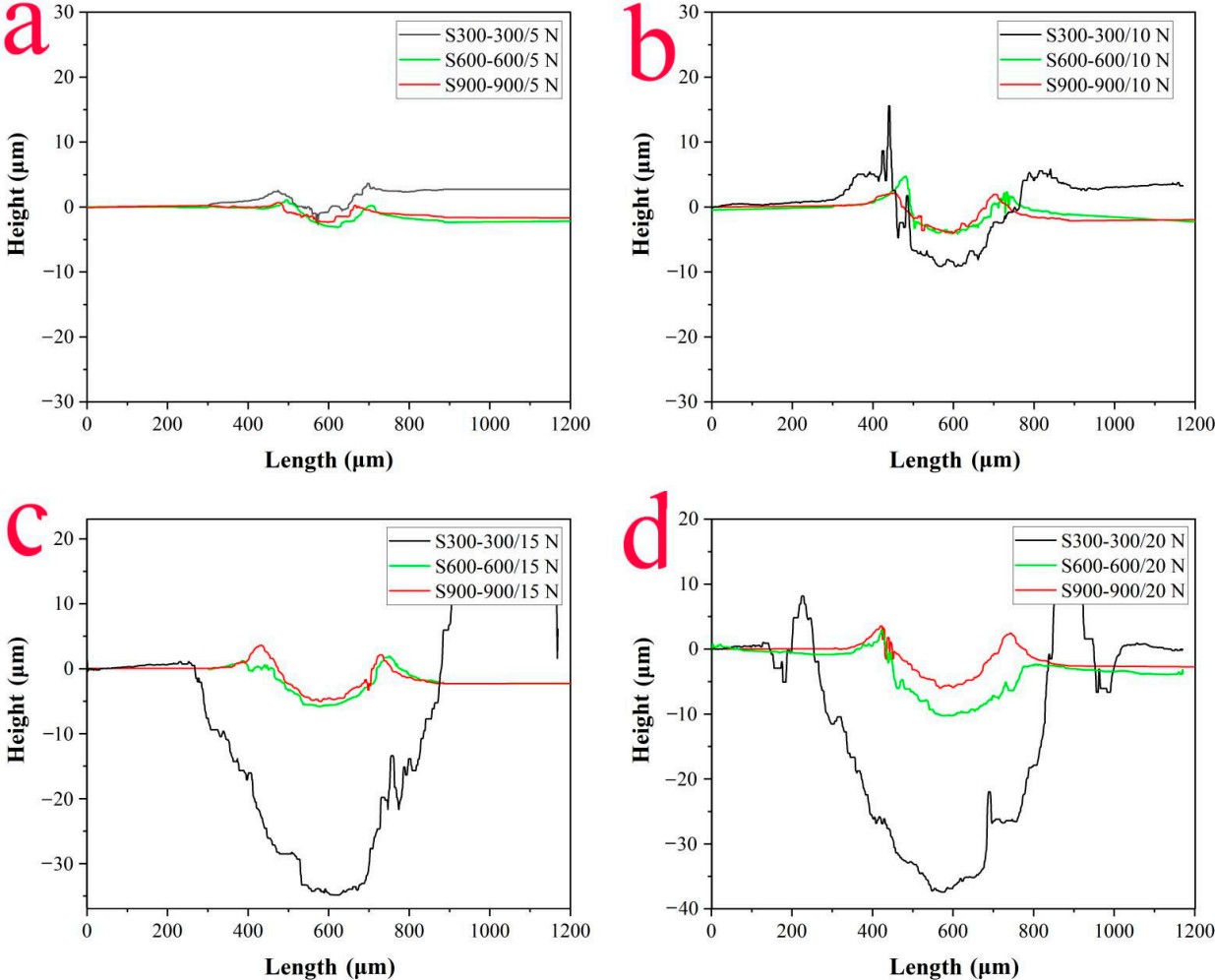

**Figure 7.** Cross-sections of the wear traces of $MoS_2$–$Ti_L$/$MoS_2$–$Ti_H$ coatings with different single layer thickness under different loads: (**a**) 5 N, (**b**) 10 N, (**c**) 15 N, (**d**) 20 N.

Figure 8 shows the wear rates of the $MoS_2$–$Ti_L$/$MoS_2$–$Ti_H$ coatings with different single layer thickness under different loads. The wear rates of all coatings were substantially lower under loads of 5 N than those under other loads. The wear rates showed an increasing trend under applied loads of up to 20 N. Under all these loads, the wear rates of the S600-600 and S900-900 coatings remained low, with the wear rates of the S900-900 coating being the lowest, thus confirming the stable lubrication properties of the S600-600 and S900-900 coatings under the test loads. However, the wear rates of the S300-300 coating considerably increased when the applied loads increased up to 20 N. The results shown in Figure 8 further proved that the S900-900 and S300-300 coatings had the best and worst wear resistance, respectively. These results are in accordance with those shown in Figures 5–7.

In conclusion, based on the above results, the adhesion properties, mechanical properties, friction behavior and wear resistance of the multilayer coatings improve with an increase in single layer thickness.

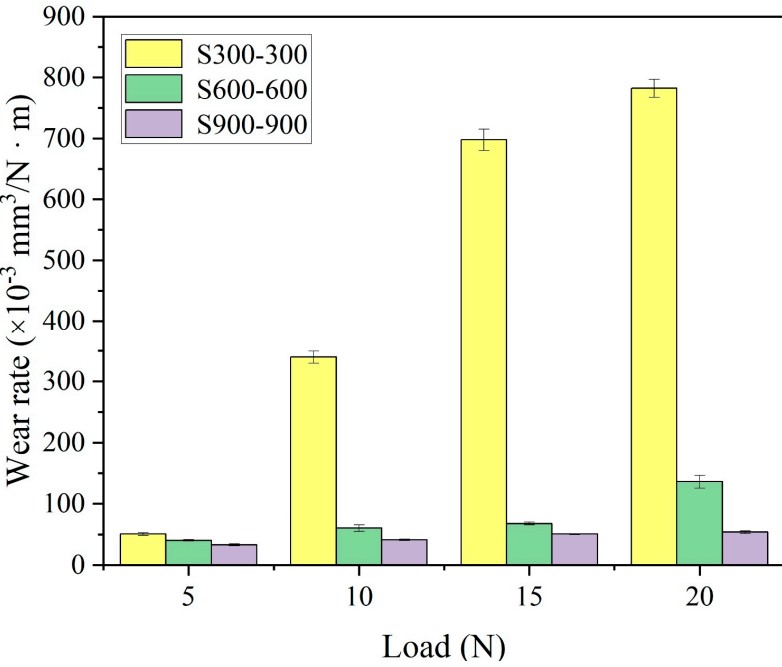

**Figure 8.** Wear rates of the $MoS_2$–$Ti_L$/$MoS_2$–$Ti_H$ coatings with different single layer thickness under different loads.

*3.2. Effect of Thickness Ratio of Single Layer on the Properties of $MoS_2$–$Ti_L$/$MoS_2$–$Ti_H$ Multilayer Coatings (S300-900, S600-600 and S900-300)*

3.2.1. Structure Characterizations of the $MoS_2$–$Ti_L$/$MoS_2$–$Ti_H$ Multilayer Coatings with Different Thickness Ratio of Single Layer

Surface and cross-sectional morphologies of the $MoS_2$–$Ti_L$/$MoS_2$–$Ti_H$ multilayer coatings with different thickness ratio of single layer are presented in Figures 9 and 10. As shown in Figure 9a–c, the surfaces of all coatings were dense and smooth, with no significant difference in the surface topography between the three images. All images in Figure 10 show the alternating layers in the multilayer structures, with each alternating layer comprising a $MoS_2$–$Ti_L$ layer and a $MoS_2$–$Ti_H$ layer, which was not evident in Figure 10. The thicknesses of the S300-900, S600-600 and S900-300 coatings shown in Figure 10 were $0.96 \pm 0.01$, $1.26 \pm 0.01$ and $1.01 \pm 0.01$ μm, respectively. Due to the multilayer structures in all coatings, the columnar growth of all coatings was inhibited, which is consistent with the results shown in Figure 10.

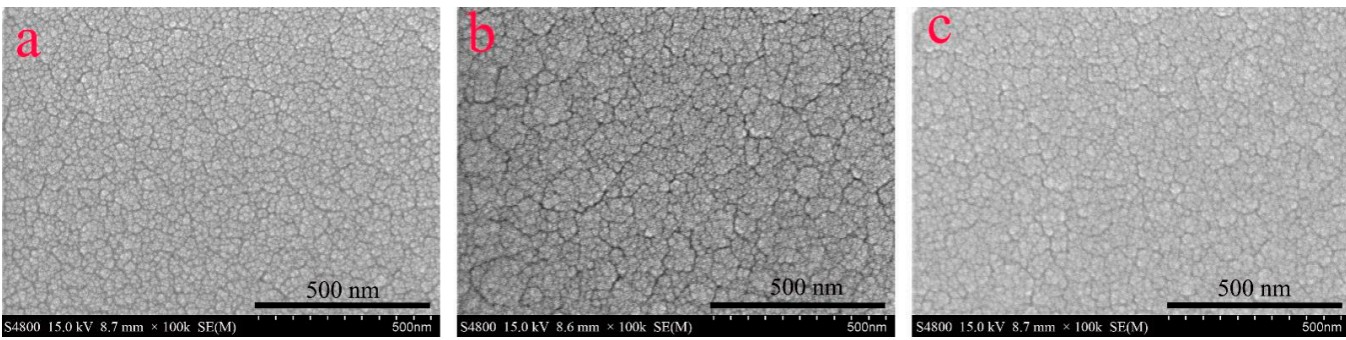

**Figure 9.** SEM surface images of $MoS_2$–$Ti_L$/$MoS_2$–$Ti_H$ coatings: (**a**) sample S300-900; (**b**) sample S600-600; (**c**) sample S900-300.

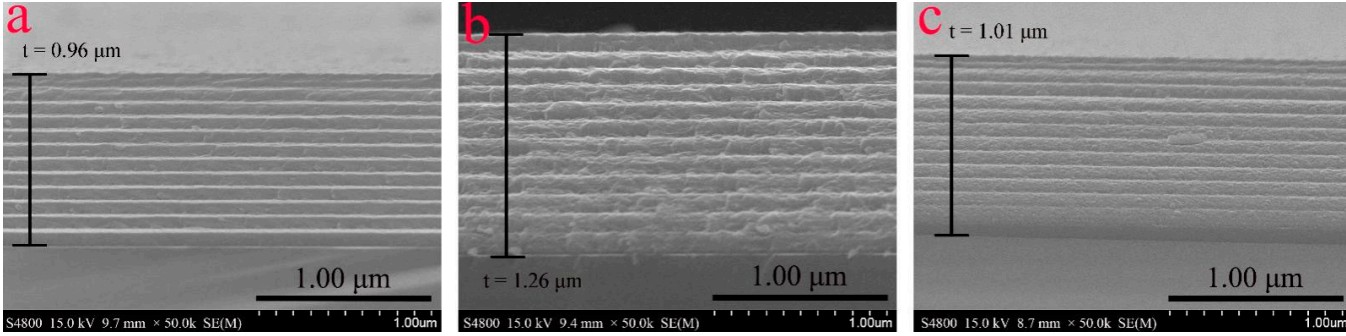

**Figure 10.** Cross-sectional SEM images of $MoS_2$–$Ti_L$/$MoS_2$–$Ti_H$ coatings: (**a**) sample S300-900; (**b**) sample S600-600; (**c**) sample S900-300.

### 3.2.2. Adhesion Properties of $MoS_2$–$Ti_L$/$MoS_2$–$Ti_H$ Multilayer Coatings with Different Thickness Ratio of Single Layer on the Steel Substrate

Adhesion properties of $MoS_2$–$Ti_L$/$MoS_2$–Ti multilayer coatings with different thickness ratio of single layer on the steel substrate were evaluated via Rockwell-C indentation. The metallographic images of the indentation craters after the test are shown in Figure 11. It is clearly observable that all coatings exhibited an area of spallation after indentation. The peeled area of the S900-300 coating in Figure 11c was found to be the smallest. Both the S300-900 and S600-600 coatings had a large area of coating spallation. The spallation of the S600-600 coating was spot-like. In addition to spot-like spallation, flaky spallation also appeared in the S300-900 coating. It can be concluded from Figure 11 that the adhesion properties of the three coatings were found to be in the order of: S900-300 > S600-600 > S300-900, which indicate that the $MoS_2$–$Ti_L$ layer had more impact on the adhesion properties of the $MoS_2$–$Ti_L$/$MoS_2$–$Ti_H$ multilayer coatings.

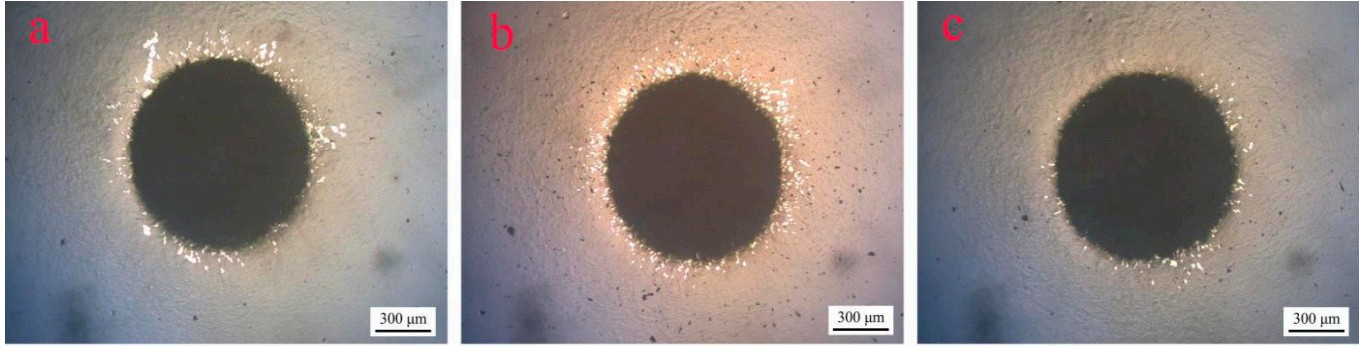

**Figure 11.** Optical micrograph of $MoS_2$–$Ti_L$/$MoS_2$–$Ti_H$ coatings after Rockwell-C indentation: (**a**) sample S300-900; (**b**) sample S600-600; (**c**) sample S900-300.

### 3.2.3. Mechanical Properties of the $MoS_2$–$Ti_L$/$MoS_2$–$Ti_H$ Multilayer Coatings with Different Thickness Ratio of Single Layer

The H and E values of the coatings with different thickness ratio of single layer are listed in Table 5. The S300-900 coating exhibited the highest hardness, followed by the S600-600 coating, with the lowest value observed for the S900-300 coating. In addition, the E moduli showed the same trend as the H values. The H/E and $H^3/E^2$ ratios are also listed in Table 5. As the deposition time of the $MoS_2$–$Ti_L$ layer increased from 300 to 900 s, the H/E and $H^3/E^2$ ratios slightly increased. The results in Table 5 indicate that the $MoS_2$–$Ti_H$ layer had more impact on the hardness and elastic moduli of the $MoS_2$–$Ti_L$/$MoS_2$–$Ti_H$ multilayer coatings.

**Table 5.** Hardness values and elastic moduli of $MoS_2$–$Ti_L$/$MoS_2$–$Ti_H$ coatings with different thickness ratio of single layer.

| Samples | Total Layers | Hardness (GPa) | Elastic Moduli (GPa) | H/E | $H^3/E^2$ |
|---------|--------------|----------------|----------------------|-----|-----------|
| S300-900 | 12 | 8.7 ± 0.1 | 155.2 ± 6.9 | 0.056 ± 0.002 | 0.027 ± 0.002 |
| S600-600 | 12 | 8.2 ± 0.5 | 144.3 ± 9.3 | 0.057 ± 0.001 | 0.027 ± 0.001 |
| S900-300 | 12 | 7.4 ± 0.4 | 121.1 ± 8.3 | 0.061 ± 0.001 | 0.028 ± 0.001 |

### 3.2.4. Tribological Properties of the $MoS_2$–$Ti_L$/$MoS_2$–$Ti_H$ Multilayer Coatings with Different Thickness Ratio of Single Layer

Figure 12 shows the friction coefficient dynamics of the $MoS_2$–$Ti_L$/$MoS_2$–$Ti_H$ multilayer coatings with different thickness ratio of single layer under different applied loads. As shown in Figure 12a–c, the friction coefficients of all coatings were similar under 5, 10 and 15 N loads, and all remained stable and low (~0.05–0.1) when the applied load was ≤15 N. However, the tribological performance of the S300-900 coating deteriorated when the applied load increased to 20 N. As exhibited in Figure 12d, the friction coefficient of the S300-900 coating suddenly increased at ~800 s. Moreover, the friction coefficients of the S900-300 and S600-600 coatings were still stable and low, evidencing their excellent tribological properties.

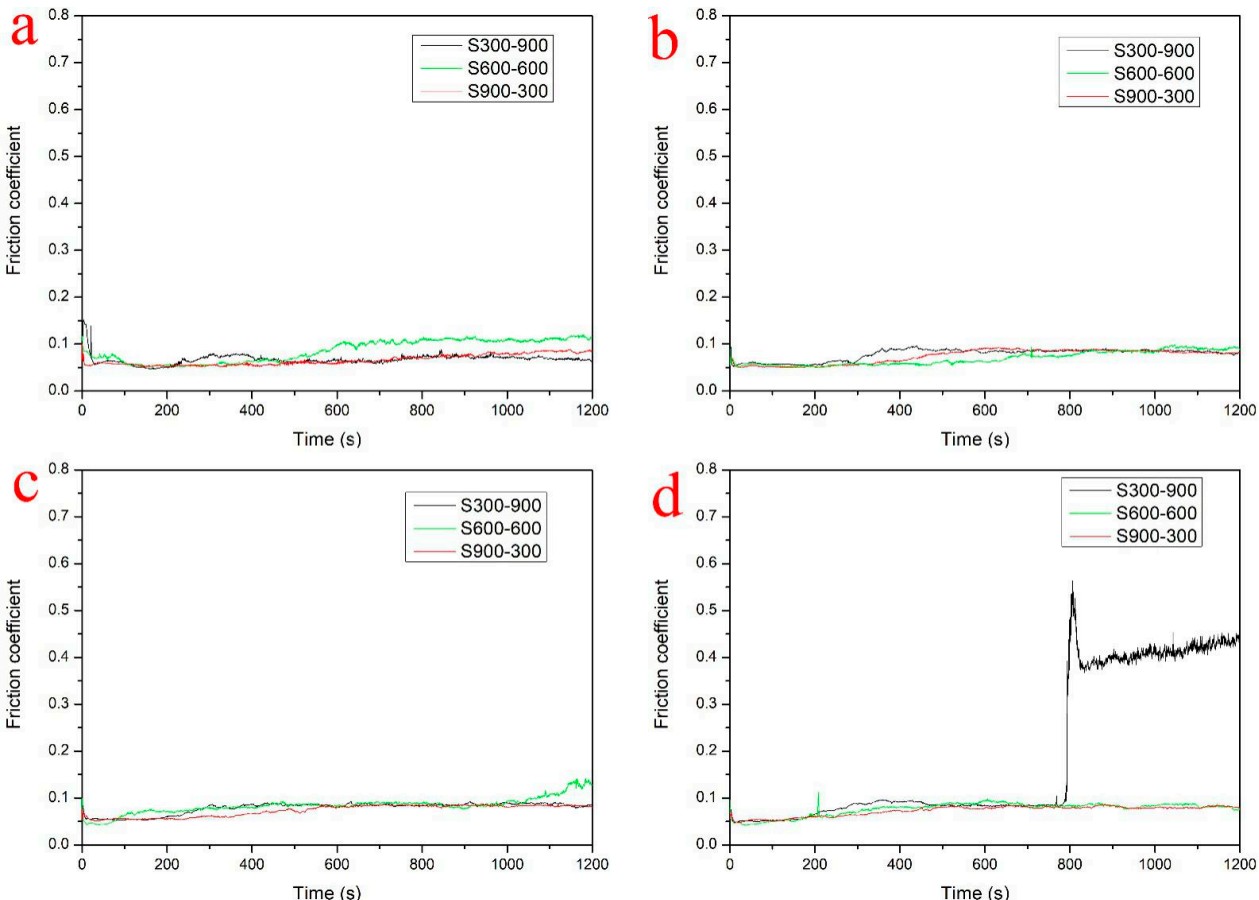

**Figure 12.** Friction coefficient dynamics of the $MoS_2$–$Ti_L$/$MoS_2$–$Ti_H$ coatings with different thickness ratio of single layer under different loads: (**a**) 5 N; (**b**) 10 N; (**c**) 15 N; (**d**) 20 N.

Figure 13 shows the wear trace morphology of the $MoS_2$–$Ti_L$/$MoS_2$–$Ti_H$ multilayer coatings with different thickness ratio of single layer under different loads, measured via confocal microscopy. For the S300-900 coating tested under 5 N load, the coating adhered

to the substrate well and stayed on the substrate, indicating that the S300-900 coating used its self-lubrication properties under a small load of 5 N. However, when the applied load increased to 10 N, the surface of the coating started to deform at the wear-trace edges, as depicted by the red rectangle shown in Figure 13d. Furthermore, when the applied load increased to 15 and 20 N, the deformation of the surface of the coating was more severe at the wear-trace edges, as depicted by the red rectangles in Figure 13g,j. Moreover, peeling of the coating can be observed from the wear trace area under 20 N; this was caused by the rapid loss of the lubrication of the coating. This finding was in accordance with the friction test results of the S300-900 coating shown in Figure 12d. Under 15 and 20 N loads, the surface of the S600-600 coating started to deform at the wear trace edges, as depicted by the red rectangles in Figure 13h,k. The S900-300 coating was firmly adhered to the substrate, with its surface slightly deformed at the wear trace edges. The S900-300 coating fully utilized its lubrication under these loads.

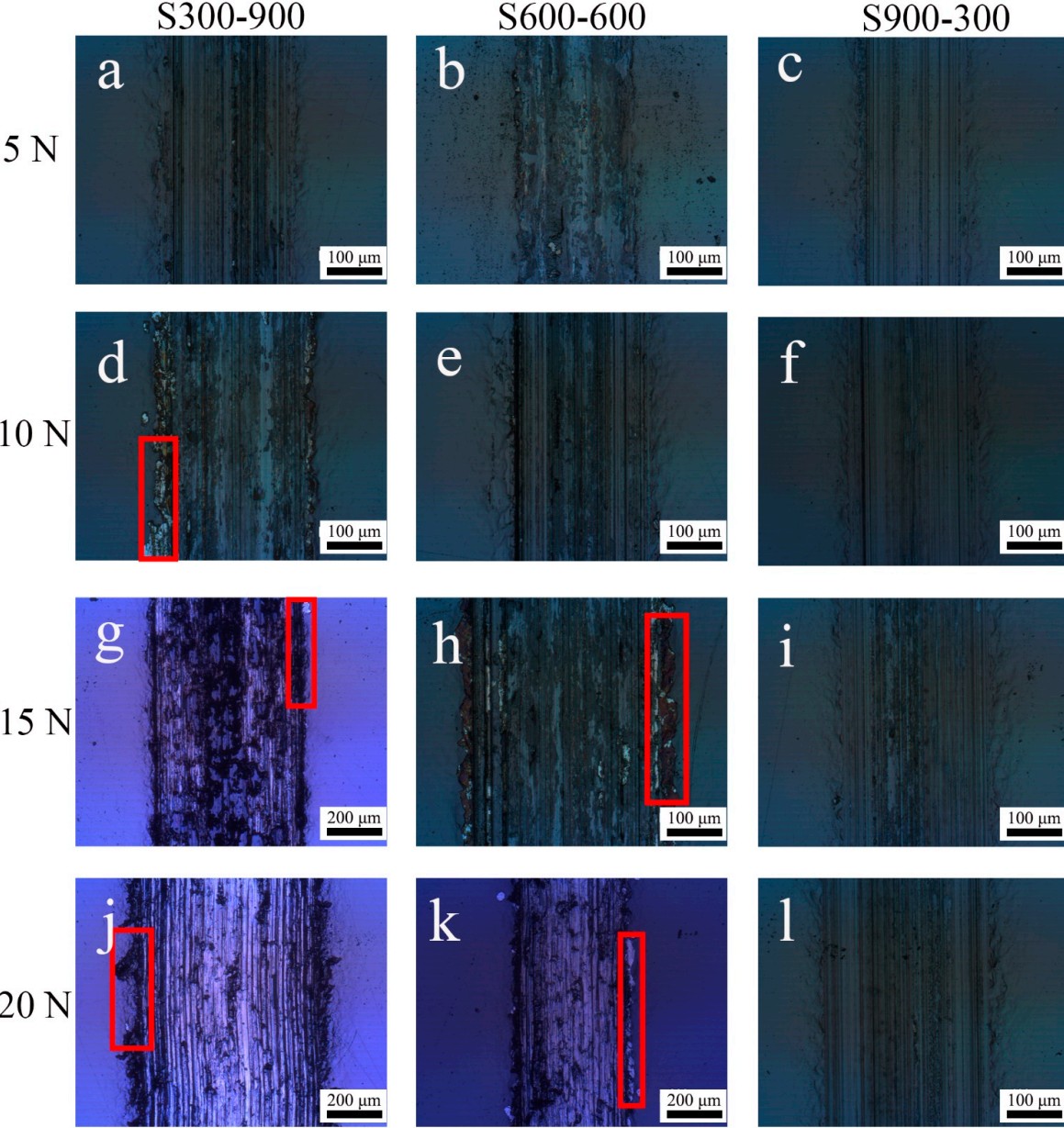

**Figure 13.** Wear trace morphology of the $MoS_2$–$Ti_L$/$MoS_2$–$Ti_H$ coating with different thickness ratio of single layer under different loads. Sample S300-900 under different loads: (**a**) 5 N, (**d**) 10 N, (**g**) 15 N, (**j**) 20 N; sample S600-600 under different loads: (**b**) 5 N, (**e**) 10 N, (**h**) 15 N, (**k**) 20 N; sample S900-300 under different loads: (**c**) 5 N, (**f**) 10 N, (**i**) 15 N, (**l**) 20 N.

Wear trace widths of the $MoS_2$–$Ti_L$/$MoS_2$–$Ti_H$ coatings with different thickness ratio of single layer (corresponding to those in Figure 13) are listed in Table 6. As shown in Table 6, the wear trace widths of all coatings increased with an increase in the applied loads from 5 to 20 N. Under each test load, the wear trace width of the S900-300 coating was the lowest, followed by that of the S600-600 coating and the largest was that of the S300-900 coating. Based on the results shown in Figures 12 and 13 and Table 6, the S900-300 and S300-900 coatings exhibited the best and worst wear resistances, respectively.

**Table 6.** Wear trace width (μm) of $MoS_2$–$Ti_L$/$MoS_2$–$Ti_H$ coatings with different thickness ratio of single layer under different loads in Figure 13.

| Samples | Loads | | | |
|---|---|---|---|---|
| | 5 N | 10 N | 15 N | 20 N |
| S300-900 | 231 ± 4 | 284 ± 8 | 581 ± 7 | 676 ± 18 |
| S600-600 | 214 ± 9 | 250 ± 5 | 391 ± 7 | 443 ± 9 |
| S900-300 | 177 ± 2 | 216 ± 4 | 265 ± 3 | 303 ± 2 |

Figure 14 shows the cross-sections of the wear traces of the multilayer coatings with different thickness ratio of single layer under different loads. As shown in Figure 14, the wear traces under 5 and 10 N were much shallower on all coatings than those obtained under 15 and 20 N. Under loads of 15 and 20 N, the wear loss of the S600-600 and S900-300 coatings was considerably lower than that of the S300-900 coating. Although the grooves of all coatings deepened when the applied loads were increased, the wear traces of the S600-600 and S900-300 coatings were shallow, whereas the wear traces of the S300-900 coating were extremely deep under loads of 15 and 20 N, again confirming that the S300-900 coating showed the worst wear resistance.

Figure 15 shows the wear rates of the $MoS_2$–$Ti_L$/$MoS_2$–$Ti_H$ multilayer coatings with different thickness ratio of single layer under different loads, which were substantially lower under loads of 5 and 10 N than those under loads of 15 and 20 N. The wear rates increased in accordance with the applied loads up to 20 N. Under these loads, the wear rates of the S600-600 and S900-300 coatings were low, with the wear rates of the S900-300 coating being the lowest, confirming the stable lubrication properties of these coatings under the test loads. However, the wear rates of the S300-900 coating substantially increased when the applied load was increased up to 15 and 20 N. The results in Figure 15 further supported that the S900-300 coating showed the best wear resistance and the S300-900 coating showed the worst wear resistance. These results are in accordance with the results shown in Figures 12–14.

In conclusion, based on the above results, the S300-900 coating exhibited the highest hardness and elastic modulus, but the worst adhesion property, friction behavior and wear resistance. Conversely, the S900-300 coating exhibited the lowest hardness and elastic modulus, but the best adhesion property, friction behavior and wear resistance. These indicate that the $MoS_2$–$Ti_H$ layer had more impact on the hardness of the $MoS_2$–$Ti_L$/$MoS_2$–$Ti_H$ multilayer coatings, whereas the $MoS_2$–$Ti_L$ layer substantially affected the adhesion property, friction behavior and wear resistance.

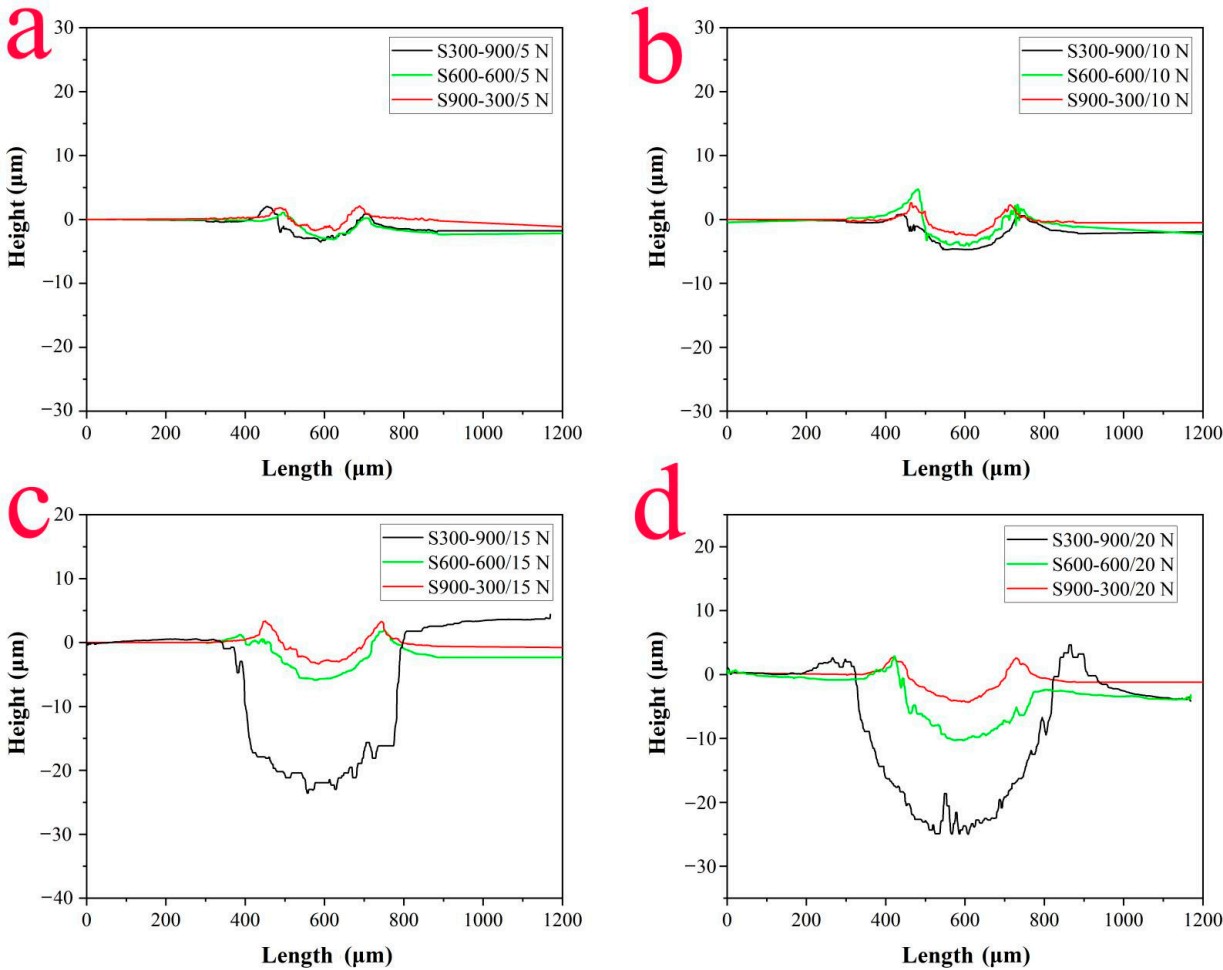

**Figure 14.** Cross-sections of the wear traces of $MoS_2$–$Ti_L$/$MoS_2$–$Ti_H$ coatings with different thickness ratio of single layer under different loads: (**a**) 5 N, (**b**) 10 N, (**c**) 15 N, (**d**) 20 N.

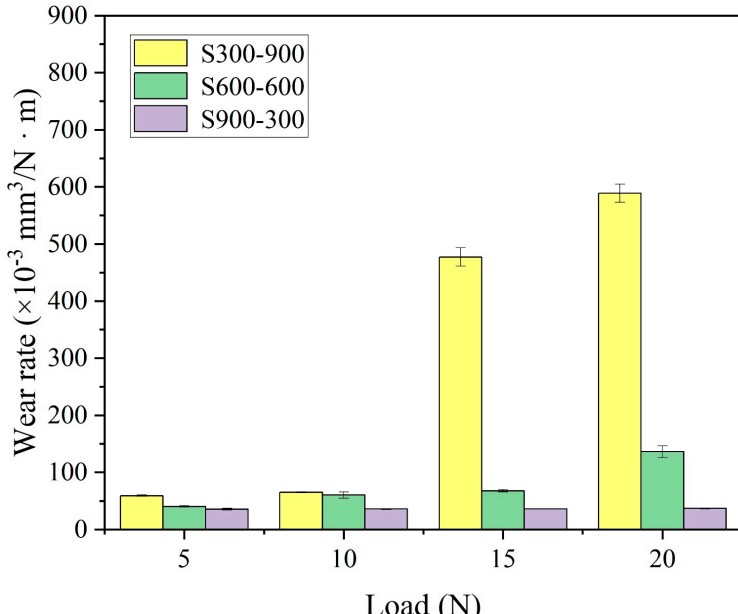

**Figure 15.** Wear rates of the $MoS_2$–$Ti_L$/$MoS_2$–$Ti_H$ coatings with different thickness ratio of single layer under different loads.

## 4. Discussion

Before discussing the effect of single layer thickness on the properties of the $MoS_2$–$Ti_L$/$MoS_2$–$Ti_H$ multilayer coatings (S300-300, S600-600 and S900-900), it should be made clear that the multilayer structure of the three coatings has two differences though the thickness ratio of $MoS_2$–$Ti_L$ to $MoS_2$–$Ti_H$ in the three coatings are all 1:1. In addition to the obvious difference in single layer thickness, the layer composition is also different, thus considering whether the layer composition of each layer affects the performance of the coatings is necessary.

The difference in the layer composition results from difference in the power of the Ti target and the deposition time of different powers. High-powered sputtering of Ti target implies that the deposited coating has high Ti content. Similarly, low-powered sputtering of Ti target implies that the deposited coating has low Ti content. The deposition time under a given power determines the layer thickness. The hardness of the multilayer coating can be calculated using the following equation [46]:

$$H = H_L V_L + H_H V_H \tag{3}$$

where $H_L$ and $H_H$ represent the hardness of the $MoS_2$–$Ti_L$ layer and $MoS_2$–$Ti_H$ layer, and $V_L$ and $V_H$ represent the volume fractions of the $MoS_2$–$Ti_L$ layer and $MoS_2$–$Ti_H$ layer. As the coating thickness is proportional to the deposition time, and the thickness fraction of $MoS_2$–$Ti_L$ can be regarded as the volume fraction of the $MoS_2$–$Ti_L$ coating, hence the deposition time ratio of $MoS_2$–$Ti_L$ to the total deposition time can be regarded as the volume fraction of the $MoS_2$–$Ti_L$ coating. Based on Equation (3), the hardness of the S300-300, S600-600 and S900-900 coatings can be calculated as follows:

$$H_{300-300} = H_L V_L + H_H V_H = H_L \times (300/600) + H_H \times (300/600) = 0.5 \times (H_L + H_H) \tag{4}$$

$$H_{600-600} = H_L V_L + H_H V_H = H_L \times (600/1200) + H_H \times (600/1200) = 0.5 \times (H_L + H_H) \tag{5}$$

$$H_{900-900} = H_L V_L + H_H V_H = H_L \times (900/1800) + H_H \times (900/1800) = 0.5 \times (H_L + H_H) \tag{6}$$

Based on the results of Equations (4)–(6) above, the hardness of the three coatings is the same, implying that the effect of the layer composition on the hardness of the studied coatings is similar if the thickness ratios of $MoS_2$–$Ti_L$ to $MoS_2$–$Ti_H$ in the three coatings are the same. It can be speculated that similar results can be applied to the adhesion property, friction behavior and wear resistance of the coatings, which indicate that the effect of the layer composition on the performance of the studied coatings is similar if the thickness ratio is the same.

### 4.1. Effect of Single Layer Thickness on the Properties of the $MoS_2$–$Ti_L$/$MoS_2$–$Ti_H$ Multilayer Coatings

Since the effect of the layer composition on the three coatings is similar, it is the single layer thickness that affects the properties of the mentioned multilayer coatings. Based on the results in Section 3.1, the adhesion properties, mechanical properties, friction behaviors and wear resistances of the multilayer coatings improve with an increase in single layer thickness (S300-300 < S600-600 < S900-900). This is mainly as a result of a combination of the following factors.

As listed in Table 1, the total deposition time of the three coatings is the same, different single layer thickness means different number of layers. As shown in Figure 3, the total thicknesses of the S300-300, S600-600 and S900-900 coatings are 0.84, 1.26 and 1.08 μm, respectively. The number of layers of the coatings is 24, 12 and 8, respectively. Therefore, the thicknesses of each layer in the S300-300, S600-600 and S900-900 coatings are 0.84/24 = 0.035, 1.26/12 = 0.105 and 1.08/8 = 0.135 μm, respectively. Different single layer thickness (or different number of layers) indicates that the effect of the multilayer interface is different. The multilayer interface increases grain boundaries, which serve as obstacles to prevent crack propagation and contribute towards improving hardness [19,37,46]. Besides,

introducing the multilayer structure significantly improved the wear resistance of $MoS_2$–Ti coating [10]. It is the positive effect of introducing the multilayer structure to improve coating performance.

However, introducing the multilayer structure has adverse effects on the coating performance. For multilayer coatings composed of different hardness layers, the internal stress increases with an increase in the number of layers [47,48]. This is because interfaces can serve as obstacles against dislocation motion, atomic rearrangement and tensile stress [37,49]. Internal stress can considerably influence the adhesion of the coating to the substrate [50]. As the internal stress increases, the hardness and critical loads of adhesion decrease [37], which may result in peeling of the coating in the friction and wear experiments [25]. This is in accordance with the result shown in Table 3 and Figures 4, 5 and 8, which indicate that the hardness, adhesion properties, friction behavior and wear resistance of the coatings deteriorate with an increase in the number of layers. That is the negative effect of the increase in the number of layers on the coating performance.

From Table 3, it can be observed that the H/E ratios of the three samples were close. As single layer thickness increased, the $H^3/E^2$ ratios slightly increased. Hardness and elastic modulus are deemed the primary properties on the basis of which the wear resistance of coatings can be evaluated [51,52]. The wear resistance of coatings can be predicted from their H/E ratio or $H^3/E^2$ ratio, which are associated with elastic strain and plastic strain to failure. Leyland et al. [51] and Tsui et al. [52] suggested that higher H/E and $H^3/E^2$ ratios implied higher wear resistance. Based on these literature studies [51,52], it confirms again that the wear resistance of the S900-900 coating is the best and that of the S300-300 coating is the worst, which is in accordance with the results shown in Figures 5 and 8.

In conclusion, the performance of the three coatings with the same thickness ratio depends on the multilayer interface since the effect of the layer composition on the studied coatings is similar. The adverse effects of the multilayer interface outweigh the benefits, evidencing that the adhesion properties, mechanical properties, friction behavior and wear resistance of the $MoS_2$–$Ti_L$/$MoS_2$–$Ti_H$ multilayer coatings improve with an increase in single layer thickness.

*4.2. Effect of Thickness Ratio of Single Layer on the Properties of the $MoS_2$–$Ti_L$/$MoS_2$–$Ti_H$ Multilayer Coating*

The S300-900, S600-600 and S900-300 coatings all have 12 layers, which indicates that the effect of the multilayer interface on all coatings is similar. Therefore, thickness ratio of single layer affects the performance of the coating. Based on Equation (3), the H values of the S300-900, S600-600 and S900-300 coatings can be calculated as follows.

$$H_{300-900} = H_L V_L + H_H V_H = H_L \times (300/1200) + H_H \times (900/1200) = 0.25 \times (H_L + 3H_H) \tag{7}$$

$$H_{600-600} = H_L V_L + H_H V_H = H_L \times (600/1200) + H_H \times (600/1200) = 0.25 \times (2H_L + 2H_H) \tag{8}$$

$$H_{900-300} = H_L V_L + H_H V_H = H_L \times (900/1200) + H_H \times (300/1200) = 0.25 \times (3H_L + H_H) \tag{9}$$

As mentioned above, using a high-power Ti target implies that the deposited coating has high Ti content. Based on previous literature results [53], the H values of the $MoS_2$–Ti coatings improve in accordance with the Ti content, which indicates that the hardness of $MoS_2$–$Ti_H$ ($H_H$) is higher than that of $MoS_2$–$Ti_L$ ($H_L$). According to the calculated Equations (7)–(9), the H values of the three coatings were found to be in the order of: S300-900 > S600-600 > S900-300, which is consistent with the results listed in Table 5. These results prove that when the coating contains the same number of multilayer interfaces, the $MoS_2$–$Ti_H$ layer plays the most important role in the hardness of the coating.

According to the Rockwell-C indentation test result in Figure 11, the adhesion properties of the three coatings were found to be in the order of: S900-300 > S600-600 > S300-900, which indicate that the $MoS_2$–$Ti_L$ layer had more impact on the adhesion properties of the coating. Additionally, this is in the same trend with their wear resistance. Özlem [25] proposed that an increased adhesion in $MoS_2$–Ta coating is beneficial to the anti-wear

performance, which is in agreement with this research that the S900-300 coating with the best adhesion has the best friction and wear resistance. This implies that the $MoS_2$–$Ti_L$ layer had more impact on the friction and wear resistance of the coating.

In addition, the wear resistance of the coatings can be predicted from the H/E and $H^3/E^2$ ratios. The H/E and $H^3/E^2$ ratios of the coatings increase in the order of: S900-300 > S600-600 > S300-900. Thus, the wear resistance of the S900-300 coating is the best and that of the S300-900 coating is the worst. This also implies that when the coating consists of the same number of multilayer interfaces, the $MoS_2$–$Ti_L$ layer plays a critical role in the friction and wear resistance of the coating.

In conclusion, the performance of three coatings with the same number of layers depends on the thickness ratio of the coating. When the effect of the multilayer interface on the studied coatings is similar, the $MoS_2$–$Ti_H$ layer has more of an effect on the hardness of $MoS_2$–$Ti_L$/$MoS_2$–$Ti_H$ coatings, whereas the $MoS_2$–$Ti_L$ layer has more of an impact on the adhesion properties, friction behavior and wear resistance of the coating. The results provide guidance for the construction of different multilayer microstructures to regulate coatings with different performance requirements.

*4.3. Analysis of Variance (ANOVA)*

Tables 7 and 8 show the analysis of variance for wear trace width and wear rate of the samples. For variation between samples, Tables 7 and 8 show that calculated value of *F* (7.64 and 5.51) is greater than critical value of F (3.26) at the 95% confidence level. Similarly for variation between loads, the tables show that the calculated *F* value (8.64 and 3.99) is greater than the critical value of *F* (3.49) at the 95% confidence level. So, the null hypothesis is not accepted and an alternative hypothesis is accepted because the calculated value of *F* is greater than the critical value of *F*. Finally, it is clear that the samples do not have the same wear trace width and wear rates for various samples and applied loads.

**Table 7.** Analysis of variance for wear trace width (μm) of the $MoS_2$–$Ti_L$/$MoS_2$–$Ti_H$ coatings from Tables 4 and 6.

| Source of Variation | Sum of Squares (SS) | Degrees of Freedom (DOF) | Mean Square (MS) | $F_{\text{calculated}}$ | $F_{\text{critical}}$ |
|---|---|---|---|---|---|
| Between samples | 375,774 | 4 | 93,943 | 7.64 | 3.26 |
| Between loads | 318,580 | 3 | 106,193 | 8.64 | 3.49 |
| Error | 147,510 | 12 | 12,292 | | |
| Total | 841,863 | 19 | | | |

**Table 8.** Analysis of variance for wear rate ($\times 10^{-3}$ $mm^3$/N·m) of the $MoS_2$–$Ti_L$/$MoS_2$–$Ti_H$ coatings from Figures 8 and 15.

| Source of Variation | Sum of Squares (SS) | Degrees of Freedom (DOF) | Mean Square (MS) | $F_{\text{calculated}}$ | $F_{\text{critical}}$ |
|---|---|---|---|---|---|
| Between samples | 561,320 | 4 | 140,330 | 5.51 | 3.26 |
| Between loads | 305,064 | 3 | 101,688 | 3.99 | 3.49 |
| Error | 305,780 | 12 | 25,482 | | |
| Total | 1172,164 | 19 | | | |

*4.4. Comparison of the Properties of $MoS_2$–Ti Coatings in this Study with Those in Other Published Articles*

As shown in Table 9, the mechanical properties of $MoS_2$–Ti coatings in other published articles are compared with those of the coatings in this paper (see Tables 3 and 5). The friction coefficient of the coating is related to the experiment conditions (such as loading force, humidity, and so on), so the friction coefficient tested under different experimental conditions is not comparable and thus it is not listed in Table 9. For most coatings listed in Table 9, hardness values are concentrated between 5 and 7 GPa. The coating with high

Ti content has the highest hardness, which can reach 9.5–9.7 GPa [8,17]. The hardness of the coating in this study varies between 7.4–8.7 GPa, which is better than that of most coatings and slightly less than that of the best in other articles. The comparison of elastic moduli among $MoS_2$–Ti coatings is similar to the situation of the hardness. The highest elastic modulus of the coating in this study is also better than that of most coatings and slightly less than that of the best. It can be seen from the above comparison that the coating studied in this research has excellent mechanical properties. Besides, H/E and $H^3/E^2$ ratios of the studied coatings are in the middle of all the coatings, which ensures that the studied coatings have a good wear resistance. It is consistent with the results shown in Figures 5 and 12 that some of the studied coatings can maintain a low friction coefficient of ~0.05 after 20 min of friction test under a large load of 20 N. This study deposited a $MoS_2$–$Ti_L$/$MoS_2$–$Ti_H$ nano-multilayer coating, which possesses excellent mechanical properties and good wear resistance.

**Table 9.** Hardness values, elastic moduli, H/E ratio and $H^3/E^2$ ratio of $MoS_2$–Ti coatings from published articles and this research.

| Author | Hardness/GPa | Elastic Moduli/GPa | H/E | $H^3/E^2$ | Note |
|---|---|---|---|---|---|
| Picas [7] | 6.2 | 160.0 | 0.039 | 0.009 | - |
| Qin [8] | 5.2–9.7 | - | - | - | A series of coatings with different Ti content |
| Zhang [10,12] | 6.7 | 101.6 | 0.066 | 0.029 | - |
| Sun [14] | 5.6 | 73.4 | 0.076 | 0.033 | Nanocomposite coating |
| | 6.8 | 82.4 | 0.083 | 0.047 | Nano-multilayer coating |
| Wang [16] | 6.4 | 108.7 | 0.059 | 0.022 | - |
| Kim [17] | 4.5–9.5 | - | - | - | A series of coatings with different Ti content |
| This study | 7.4–8.7 | 121.1–155.2 | 0.056–0.061 | 0.025–0.028 | A series of coatings prepared by different modulation periods |

- indicates that data is not provided by the article.

## 5. Conclusions

In this study, the effect of different modulation periods on the mechanical and tribological performance of multilayer coatings was studied. The performance of the studied multilayer coatings was found to be dependent on modulation periods of single layer thickness and thickness ratio. Overall, when the thickness ratio of $MoS_2$–$Ti_L$/$MoS_2$–$Ti_H$ single layer was fixed with different numbers of layers, the adverse effects of the interface outweighed the beneficial effects, thus implying that the adhesion properties, mechanical properties, friction behavior and wear resistance of the $MoS_2$–$Ti_L$/$MoS_2$–$Ti_H$ multilayer coatings improve with an increase in single layer thickness. When the effect of the multilayer interfaces on the studied coatings was similar with the same number of layers, the $MoS_2$–$Ti_H$ layer had more of an effect on the hardness of the $MoS_2$–$Ti_L$/$MoS_2$–$Ti_H$ coatings, whereas the $MoS_2$–$Ti_L$ layer had more of an impact on the adhesion properties, friction behavior and wear resistance of the coatings.

**Author Contributions:** Data curation, P.Y. and C.L.; funding acquisition, V.L.; investigation, P.Z. and P.Y.; methodology, P.Z. and V.L.; writing—original draft, P.Y.; writing—review & editing, P.Z., C.L., T.Y., J.W., M.H., T.W. and Y.F.; supervision, T.Y., J.W., M.H., T.W. and Y.F. All authors have read and agreed to the published version of the manuscript.

**Funding:** The authors are thankful for the financial support of the Zhejiang Provincial Natural Science Foundation of China under grant No. LQ17E020001 and No. LTZ20E020001, Public Welfare Projects of Science and Technology Department of Zhejiang Province under grant No. 2016C37049, the Foundation of Zhejiang Educational Committee under grant No. Y201738304, and the National Key Research and Development Plan under grant No. 2018YFB1107305.

**Institutional Review Board Statement:** Not applicable.

**Informed Consent Statement:** Not applicable.

**Data Availability Statement:** Data is available from the Dryad Digital Repository. Data can be accessed through URL: https://datadryad.org/stash/share/k4b8Pv_yUOlE9TwD_AVJzNeLZJpAY1 SSOyjgIxIgGNA (accessed on 27 August 2021).

**Conflicts of Interest:** The authors declare no conflict of interest.

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
