# Peer review of "Effect of Modulation Periods on the Mechanical and Tribological Performance of MoS2–TiL/MoS2–TiH Multilayer Coatings"

_coatings, doi:10.3390/coatings11101230_

Round 1

Reviewer 1 Report

You have done a very interesting research. I like very much the review in the introduction about the use of different metals with MoS2. The manuscript in general is fine but some things need to be fixed:

  1. Depositions rate of 4 hours indicated in Tables 1 and 2 don’t seem to be very realistic from an industrial production point of view. Discuss in the manuscript how these times could be shortened. If you propose higher deposition rates, remember that the quality of the coatings change.
  2. Indicate in the text the experimental conditions for the nanoindentation: quasi-static or dynamic indentation, kind of indenter (cubic, Berkovich, etc), maximum applied load, loading/unloading rate, time at maximum load. Also indicate how many indents were done per sample to give an error to each H and E value. Indicate if you use the Oliver and Phar theory for the calculation of H and E. 
  3. Indicate in the text the maximum penetration depth for each kind of coating. Also explain how the substrate influences the E and H values.
  4. Indicate in the text the Poisson’s ratio you have chosen for each coating.
  5. Indicate errors in Table 3 for all data.
  6. How the values in Table 3 compares to already published data? Make a discussion in your text.
  7. Please, indicate the initial Hertzian contact pressures in your tribological experiments. Also comment how these values represent (or not) real applications/situations.
  8. Indicate how many tribological tests were performed per sample, and how are the dispersion values of friction and wear when analyzing the repeated experiments.
  9. Please, indicate adhesion values for your coatings on the metal substrates. If you cannot perform scratch tests, at least make a qualitative study with Rockwell-C Adhesion Indentation tests.

Reviewer 2 Report

Abstract:

  1. The motivation of your study should be better highlighted in this part. Thi big question "why" should be answered synthetically.

Introduction:

  1. The importance of your research should be expressed explicitly. This sentence is not enough: " However, the effects of modulation periods
    such as single layer thickness and thickness ratio on the mechanical and tribological performance of the MoS2–TiL/MoS2–TiH multilayer coatings were not detailed in our previous study, and thus requires further investigation."

Materials and methods:

  1. Section named "Experimental details" is confusing. I would strongly suggest to use conventional naming, e.g. materials and methods.
  2. Why silicon and stainless steel were chosen as a substrate?
  3. How was "mirror finish" evaluated? Any roughness measurements prior to the deposition? What kind of processing were used to achieve the finish. My concern is that internal stress could be induced by certain manufacturing techniques which may affect the performance of deposited coating.
  4. An image of the stand used for deposition should be provided.
  5. The explanation of why the authors used the depositing parameters should be clearly provided.
  6. How many measurements were taken when it comes for hardness/Young's modulus/tribological parameters etc?
  7. What kind of statistical analysis was used to determine if the assumed controlleable factors are relevant? What kind of statistical tests were used to compare the results?

Results

  1. Quality of figure 5 is bad.
  2. Some of the methods description should be stated in the previous section. I would like to see the actual results here.
  3. The results are here compared without any statistical tools. This is a serious flaw of this research.
  4. My other concern is that some images are similar or almost identical - Figure 4k and 10k or Figure 2b and 8b. This has to be carefully addressed here!

Discussion part is non-existent. The author merely describe their result but do not produce anything new which would contribute to better understanding of the phenomena occurring in the process they deal with. The reference and discussing their results with other work is also needed.

Reviewer 3 Report

The manuscript entitled: Effect of modulation periods on the mechanical and tribological performance of MoS2-TiL/MoS2-TiH multilayer coatings, where the effect of two modulation periods on the mechanical and tribological properites of the MoS2 based multilayer coatings is studied. I have the following concerns with the manuscript:

  • A strong scientific discussion correlating the coatings and the tribological and mechanical performance is missing. The influence of the number of layers on these properties should be correlated with in-depth scientific arguments.
  • Figure 1 both scale bar and the features on the surface are hardly visible. Higher resolution images with better scale bar is preferred.
  • The values 0.89, 1.28, 1.08 microns measured from Fig. 2 may not be true, especiall the last two coating thicknesses of 1.28 and 1.08 microns. Please double check.
  • Unit of friction coefficient is missing in Figs. 3 and 9. If there is no unit mention as (no unit).
  • In the scale bar in Figs. 4 and 10 space should be introduced between number and unit.
  • Table 4 - 5N should be written as 5 N and so on..
  • Figs. 5 and 11 - legend on Z-axis is missing.
  • Figs. 5 and 11 are not of publishable quality. Please modify the images.
  • Fig. 6 - Error bars should be introduced.
  • Equation numbers for all the equations should be introduced in a sequential order.
  • High resolution images may be introduced in Fig. 7.
  • Again the thickness values measured from Fig. 8 seems to have some errors. Please double check!

Round 2

Reviewer 2 Report

Still the authors did not provide the appropriate statistical test to determine if the results are statistically different. This concerns in particular data presented in Table 4 and 6 as well as Figures 8 and 15. ANOVA if applicable is highly recommended here.
